# The Spatiotemporal Pattern of Decoupling Transport CO$_2$ Emissions from Economic Growth across 30 Provinces in China

**Ji Zheng [1,2], Yingjie Hu [3], Suocheng Dong [1,2,*] and Yu Li [1,2,*]**

[1] Institute of Geographic Sciences and Natural Resources Research, Chinese Academy of Sciences, Beijing 100101, China; zhengj.13s@igsnrr.ac.cn

[2] University of Chinese Academy of Sciences, Beijing 100049, China

[3] College of Land Science and Technology, China Agricultural University, Beijing 100193, China; Yingjiehu@cau.edu.cn

[*] Correspondence: dongsc@igsnrr.ac.cn (S.D.); liy@igsnrr.ac.cn (Y.L.); Tel.: +86-10-64889430 (S.D.); +86-10-64889107 (Y.L.)

**Abstract:** Since 2005, China has become the largest emitter of CO$_2$. The transport sector is a major source of CO$_2$ emissions, and the most rapidly growing sector in terms of fuel consumption and CO$_2$ emissions in China. This paper estimated CO$_2$ emissions in the transport sector across 30 provinces through the IPCC (International Panel on Climate Change) top-down method and identified the spatiotemporal pattern of the decoupling of transport CO$_2$ emissions from economic growth during 1995 to 2016 by the modified Tapio's decoupling model. The CO$_2$ emissions in the transport sector increased from 103.10 million ton (Mt) in 1995 to 701.04 Mt in 2016. The year, 2005, was a turning point as the growth rate of transport CO$_2$ emissions and the intensity of transport CO$_2$ emissions declined. The spatial pattern of transport CO$_2$ emissions and its decoupling status both exhibited an east-west differentiation. Nearly 80% of the provinces recently achieved decoupling, and absolute decoupling is beginning to take place. The local practices of Tianjin should be the subject of special attention. National carbon reduction policies have played a significant role in achieving a transition to low-carbon emissions in the Chinese transport sector, and the integration of multi-scale transport CO$_2$ reduction policies will be promising for its decarbonisation.

**Keywords:** climate change; indicator of decoupling; negative environmental externalities of the transport sector; transition to low carbon economy

## 1. Introduction

The reduction of CO$_2$ emissions is a global environmental challenge [1]. The United Nations identified 'take urgent action to combat climate change and its impacts' as one of the goals to achieve sustainable development across the globe [2]. According to the International Energy Agency, the transport sector accounts for 28.81% of total energy consumption in 2017 [3] and 25% of total CO$_2$ emissions in 2016 [4]. The growth of CO$_2$ emissions is led by developing countries experiencing rapid economic growth [5]. In many developing countries, CO$_2$ emissions in the transport sector have attracted great attention because of their high contribution and unprecedented increase in scale and speed [1,6]. China has become the largest country of CO$_2$ emissions since 2005 [7]. The transport sector is a major source of CO$_2$ emissions and the most rapidly growing sector in terms of fuel consumption and CO$_2$ emissions in China [8,9]. The key challenge for China, as well as other developing countries, is what can be done to reduce CO$_2$ emissions in the transport sector while achieving economic development [10], especially in the aspect of environmental governance [11].

Decoupling $CO_2$ emissions in the transport sector from economic growth is the key to providing a local practical solution to low-carbon development. In 2000, this concept was first introduced by Zhang et al. [12]. In 2012, the concept of decoupling was developed as an indicator by the OECD (Organization for Economic Co-operation and Development, OECD) [13]. Vehmas et al. [14] and Tapio [15] then made great progress in decoupling: Vehmas et al. [14] constructed a framework that explained different aspects of decoupling and Tapio [15] proposed the elastic decoupling model. The decoupling of $CO_2$ emissions from economic growth subsequently became a hot topic [16–19], since economic growth is desirable, but $CO_2$ emissions are not. Three decoupling indicators have been employed most frequently in previous research to reveal the relationship between $CO_2$ emissions and economic growth: (1) Indicator *DO*, introduced by the OECD [13], represents the ratio of environmental pressure to economic growth for a given research year in relation to a base year; (2) indicator *DT*, introduced by Tapio [15], denotes the emission-to-economic activity elasticity; and (3) indicator *DL*, introduced by Lu et al. [20] and developed based on the IPAT (Human Impact, Population, Affluence and Technology) framework [21], represents the decreasing rate of emissions' intensity. Indicator *DT* can distinguish between absolute decoupling and relative decoupling, which indicator *DO* cannot, and it can also distinguish the 'absolute decoupling' between an economy's recession and growth periods, which indicator *DL* cannot. However, the existing decoupling indicators cannot capture the relationship between $CO_2$ emissions and economic growth in terms of per capita. Placing the blame on developing countries for causing massive $CO_2$ emissions worldwide is questionable from both the historical and per capita perspective [6]. In addition, important effects of individual characteristics and behaviors on transport $CO_2$ emissions have been observed [22,23]. The decoupling indicator in terms of per capita is also especially important for large developing countries, such as China.

The transport sector has recently attracted much attention in decoupling research. Most existing research has focused on the national level, regional level, and one or several provinces or cities. Loo and Banister [6] extended the discussion on the decoupling of environmental and social issues in the transport sector from economic growth by examining absolute and relative decoupling in a strong and a weak version. Wu et al. [3] analyzed the decoupling states of $CO_2$ emissions in the Chinese transport sector from the perspective of fuel types and found a negative or a non-existent decoupling state accounted for 72.2% during the study period. Wang et al. [10] estimated the decoupling elasticity between economic growth and $CO_2$ emissions in the transport sector in China and analyzed the key factors driving Chinese transport $CO_2$ emissions during 2000 to 2015. Due to the imbalanced regional development in China [24,25], it is necessary to undertake a comparative study of the decoupling of transport $CO_2$ emissions from economic growth at the regional and provincial level. Zhu and Li [5] analyzed the decoupling of $CO_2$ emissions in the transport sector from economic growth in the Beijing-Tianjin-Hebei (BTH) area of China. The changes in transport $CO_2$ emissions and economic growth were not synchronized during 2005 to 2013 in BTH. Guo et al. [1] analyzed transport $CO_2$ emission patterns in China at both regional and provincial levels and found significant regional disparities in transport $CO_2$ emissions in China. In China, national $CO_2$ reduction targets are assigned to the provincial level, and the mitigation policies at the provincial level are more pertinent and flexible. Therefore, the decoupling analysis of transport $CO_2$ emissions at the provincial level has attracted much attention. Zhao et al. [7] found that the year, 2005, was a turning point in the transport $CO_2$ emissions' decoupling level in Guangdong province, China. Dong et al. [26] took Xinjiang as a case to conduct decoupling analysis on the decoupling of transport $CO_2$ emissions from economic growth, and they found a fluctuating decoupling pattern from 1990 to 2014. Wang et al. [27] conducted research on the decoupling of carbon emissions from economic growth in the transportation sector in Jiangsu and proposed that transportation has made achievements in reducing $CO_2$ emissions since 2010. However, the inter-regional comparability of previous research results is relatively poor due to different decoupling indicators and decoupling state typology. Wu et al. [28] reviewed the mitigation policies related to on-road transportation emissions in China and proposed that most of the previous transport $CO_2$ mitigation policies were at the national level. It is necessary to research the spatiotemporal pattern

of the decoupling of transport $CO_2$ emissions from economic growth at the provincial level in China to identify provinces with good local practices and to provide a scientific basis for the formulation of appropriate transport $CO_2$ mitigation policies, taking into consideration local factors.

In order to address the abovementioned research shortcomings, this study developed an extended decoupling indicator in terms of per capita, and explored the spatiotemporal pattern of the decoupling of transport $CO_2$ emissions from economic growth across 30 provinces in China during the period 1995–2016. This study presents three major contributions by: (1) Evaluating the $CO_2$ emissions in the transport sector for 30 provinces from 1995 to 2016; (2) uncovering the spatiotemporal pattern of transport $CO_2$ decarbonisation at the provincial level; and (3) identifying good local practices to achieve decarbonisation in the transport sector. The remaining structure of this paper is as follows: The research methodology is described in Section 2; the results are presented and discussed in Section 3; and Section 4 summarizes the conclusions and proposes transport $CO_2$ reduction policies.

## 2. Methods and Data Collection

### 2.1. Data Description

The provincial energy consumption data used to calculate the $CO_2$ emissions in the transport sector in China from 1995 to 2016 are collected from the China Energy Statistical Yearbook [29]. Ten types of fuels are considered in this study: Raw coal, cleaned coal, coke, crude oil, gasoline, kerosene, diesel oil, fuel oil, liquefied petroleum gas (LPG), and natural gas. According to the Classification of National Economic Industries (GB/T 4754-2017), the transportation, warehousing, and postal service includes railway transportation industry, road transportation industry, ship transportation industry, air transportation industry, pipeline transportation industry, multimodal transport and transport agent industry, handling and warehousing industry, and postal industry. The GDP and population data are collected from the China Statistical Yearbook [30]. To eliminate the effect of price fluctuations, GDP and GDP per capita values are converted into the constant price in 2016 from the current prices using the consumer price index (CPI). Due to data limitations in Taiwan, Hong Kong, Macao, and Tibet, these areas are excluded in this study. Data of Chongqing is from 1997 to 2016 since it was established in 1997. Data of Fujian from 1997 to 1998 and Ningxia from 2000 to 2002 are absent in the statistical yearbook.

### 2.2. Calculation of $CO_2$ Emissions in the Transport Sector

According to the IPCC (International Panel on Climate Change, IPCC) method of greenhouse gas emission inventories [31], the "top-down" approach is applied to calculate the $CO_2$ emissions of the transport sector of 30 provinces in China from 1995 to 2016. The method can be expressed as the following formula:

$$C = \sum_i E_i \times F_i \times K_i, \tag{1}$$

where $C$ denotes the total $CO_2$ emissions of the transport sector; $E_i$ denotes the consumption of the ith fuel; $F_i$ denotes the conversion coefficient of standard coal of the ith fuel; and $K_i$ denotes the $CO_2$ emission factor of the ith fuel. The conversion coefficients of standard coal of diverse fuels, shown in Table 1, are based on the China Energy Statistical Year Book [29].

**Table 1.** The conversion coefficients of standard coal of diverse fuels [1].

| Fuel | $F_i$ (tce/t) |
|------|------|
| Raw coal | 0.7143 |
| Cleaned coal | 0.9000 |
| Coke | 0.9714 |
| Crude oil | 1.4286 |
| Gasoline | 1.4714 |
| Kerosene | 1.4714 |

**Table 1.** *Cont.*

| Fuel | $F_i$ (tce/t) |
| --- | --- |
| Diesel oil | 1.4571 |
| Fuel oil | 1.4286 |
| Liquefied petroleum gas (LPG) | 1.7143 |
| Natural gas | 1.2000 tce/$10^4$ m$^3$ |

[1] The tce denotes ton of standard coal equivalent; $F_i$ denotes the conversion coefficients of standard coal of diverse fuels based on China Energy Statistical Year Book; the conversion coefficient of $CO_2$ emissions of standard coal is 2.204 kg/kg.

### 2.3. Tapio Decoupling Model

The decoupling model proposed by Tapio [15] is widely used to analyze the relationship between economic growth and negative environmental externalities [32,33], especially $CO_2$ emissions in the transport sector [34]. The decoupling index from year T-j to year T can be calculated as follows:

$$D^T_{T-j} = \frac{\Delta C^T_{T-j}}{\Delta GDP^T_{T-j}}, \tag{2}$$

where $D^T_{T-j}$ denotes the decoupling index from year T-j to year T; $\Delta C^T_{T-j}$ denotes the $CO_2$ emissions change percentage from year T-j to year T; and $\Delta GDP^T_{T-j}$ denotes the economic growth change percentage from year T-j to year T. $\Delta C^T_{T-j}$ and $\Delta GDP^T_{T-j}$ can be calculated as follows:

$$\Delta C^T_{T-j} = \frac{\left(C^T - C^{T-j}\right)}{C^{T-j}}, \tag{3}$$

$$\Delta GDP^T_{T-j} = \frac{\left(GDP^T - GDP^{T-j}\right)}{GDP^{T-j}}. \tag{4}$$

The relationship between GDP and $CO_2$ emissions in the transport sector at a given time can be calculated as follows:

$$CE^T = \frac{C^T}{GDP^T}, \tag{5}$$

where $CE^T$ denotes the transport the $CO_2$ emissions intensity per unit GDP.

### 2.4. Decoupling Indicator in Terms of Per Capita and Decoupling Typology

To better understand and reduce the transport $CO_2$ emissions in developing countries while they strive for economic development, the decoupling index in terms of per capita is proposed as follows:

$$DP^T_{T-j} = \frac{\Delta CP^T_{T-j}}{\Delta GDPP^T_{T-j}}, \tag{6}$$

where $DP^T_{T-j}$ denotes the decoupling index in terms of per capita from year T-j to year T; $\Delta CP^T_{T-j}$ denotes the transport $CO_2$ emissions per capita change percentage from year T-j to year T; and $\Delta GDPP^T_{T-j}$ denotes the GDP per capita change percentage from year T-j to year T.

The degrees of decoupling between transport $CO_2$ emissions and GDP are divided into 8 types [6], and the typology is shown in Table 2.

**Table 2.** The typology of changing relationships between transport $CO_2$ emissions and GDP.

| | Transport $CO_2$ Emissions or Per Capita Transport $CO_2$ Emissions | |
| --- | --- | --- |
| | Decrease ($\Delta C_{T-j}^T < 0$ or $\Delta CP_{T-j}^T < 0$) | Increase ($\Delta C_{T-j}^T > 0$ or $\Delta CP_{T-j}^T > 0$) |
| **Growth** ($\Delta GDP_{T-j}^T > 0$ or $\Delta GDPP_{T-j}^T > 0$) | IIIa<br>$CE^T$ falls<br>$\lvert\Delta GDP_{T-j}^T\rvert > \lvert\Delta C_{T-j}^T\rvert$ or $\lvert\Delta GDPP_{T-j}^T\rvert > \lvert\Delta CP_{T-j}^T\rvert$<br>$\lvert D_{T-j}^T\rvert < 1$ or $\lvert DP_{T-j}^T\rvert < 1$, negative<br>Decoupling: absolute; weak | Ia<br>$CE^T$ increases<br>$\lvert\Delta GDP_{T-j}^T\rvert \le \lvert\Delta C_{T-j}^T\rvert$ or $\lvert\Delta GDPP_{T-j}^T\rvert \le \lvert\Delta CP_{T-j}^T\rvert$<br>$\lvert D_{T-j}^T\rvert \ge 1$ or $\lvert DP_{T-j}^T\rvert \ge 1$, positive<br>Coupling: relative; strong |
| | IIIb [2]  *(green)*<br>$CE^T$ falls<br>$\lvert\Delta GDP_{T-j}^T\rvert \le \lvert\Delta C_{T-j}^T\rvert$ or $\lvert\Delta GDPP_{T-j}^T\rvert \le \lvert\Delta CP_{T-j}^T\rvert$<br>$\lvert D_{T-j}^T\rvert \ge 1$ or $\lvert DP_{T-j}^T\rvert \ge 1$, negative<br>Decoupling: absolute; strong | Ib<br>$CE^T$ falls<br>$\lvert\Delta GDP_{T-j}^T\rvert > \lvert\Delta C_{T-j}^T\rvert$ or $\lvert\Delta GDPP_{T-j}^T\rvert > \lvert\Delta CP_{T-j}^T\rvert$<br>$\lvert D_{T-j}^T\rvert < 1$ or $\lvert DP_{T-j}^T\rvert < 1$, positive<br>Decoupling: relative; weak |
| **Decline** ($\Delta GDP_{T-j}^T < 0$ or $\Delta GDPP_{T-j}^T < 0$) | IVa<br>$CE^T$ increases<br>$\lvert\Delta GDP_{T-j}^T\rvert > \lvert\Delta C_{T-j}^T\rvert$ or $\lvert\Delta GDPP_{T-j}^T\rvert > \lvert\Delta CP_{T-j}^T\rvert$<br>$\lvert D_{T-j}^T\rvert < 1$ or $\lvert DP_{T-j}^T\rvert < 1$, positive<br>Coupling: relative; weak | IIa<br>$CE^T$ increases<br>$\lvert\Delta GDP_{T-j}^T\rvert > \lvert\Delta C_{T-j}^T\rvert$ or $\lvert\Delta GDPP_{T-j}^T\rvert > \lvert\Delta CP_{T-j}^T\rvert$<br>$\lvert D_{T-j}^T\rvert < 1$ or $\lvert DP_{T-j}^T\rvert < 1$, negative<br>Coupling: absolute; weak |
| | IVb<br>$CE^T$ falls<br>$\lvert\Delta GDP_{T-j}^T\rvert \le \lvert\Delta C_{T-j}^T\rvert$ or $\lvert\Delta GDPP_{T-j}^T\rvert \le \lvert\Delta CP_{T-j}^T\rvert$<br>$\lvert D_{T-j}^T\rvert \ge 1$ or $\lvert DP_{T-j}^T\rvert \ge 1$, positive<br>Decoupling: relative; strong | IIb [2]  *(dark gray)*<br>$CE^T$ increases<br>$\lvert\Delta GDP_{T-j}^T\rvert \le \lvert\Delta C_{T-j}^T\rvert$ or $\lvert\Delta GDPP_{T-j}^T\rvert \le \lvert\Delta CP_{T-j}^T\rvert$<br>$\lvert D_{T-j}^T\rvert \ge 1$ or $\lvert DP_{T-j}^T\rvert \ge 1$, negative<br>Coupling: absolute; strong |

*(Left side label spanning vertically: GDP or GDP per capita)*

[2.] The most desirable decoupling category is in green, while the most undesirable decoupling category is in dark gray.

Both economic growth and minimized transport $CO_2$ emissions are desirable, but these two goals are not always compatible. For developing countries, economic development is one of the most urgent goals. Attention should be paid to the provinces that belong to the most desirable or undesirable categories, in order to learn or avoid their practices. This can serve as a reference for other Chinese provinces and developing countries for the formulation of more effective and reasonable policies to mitigate $CO_2$ emissions in the transport sector.

## 3. Results and Discussion

### 3.1. Spatiotemporal Patterns of $CO_2$ Emissions in the Transport Sector

3.1.1. Total Amount and the Energy Structure of Transport $CO_2$ Emissions

Figure 1 shows the national $CO_2$ emissions in the transport sector and their contribution to the total national $CO_2$ emissions in China from 1995 to 2016. The national transport $CO_2$ emissions increased 6.80-fold, from 103.10 million tons (Mt) in 1995 to 701.04 Mt in 2016. The average annual growth rate of national $CO_2$ emissions in the transport sector was 9.56%, more than the average annual growth rate of $CO_2$ emissions (0.78%) in the agriculture sector during 1997 to 2014, as demonstrated by Luo et al. [35] From 1995 to 2000, national traffic $CO_2$ emissions increased continuously at the rate of 7.25%. Transport $CO_2$ emissions increased sharply with a growth rate of 19.54% from 2000 to 2005. After 2005, the growth rate of national transport $CO_2$ emissions noticeably slowed down. From 2005 to 2010, the average annual growth rate of national transport $CO_2$ emissions was 9.21%. The average annual growth rate was 3.97% during 2010 to 2016, even lower than during the period of 1995 to 2000. In addition, during 2012 to 2013, transport $CO_2$ emissions showed a marked decrease. From 1995 to 2016, the contribution of transport $CO_2$ emissions to the total $CO_2$ emissions increased from 4.47% to 9.10%. In the first five years of the study period, the proportion of transport $CO_2$ emissions relative to the total $CO_2$ emissions increased rapidly from 4.47% to 7.61%. During 2000 to 2010, the contribution of the transport sector fluctuated around 7.5%. After 2010, the proportion of $CO_2$ emissions from the transport sector showed a slowly increasing trend.

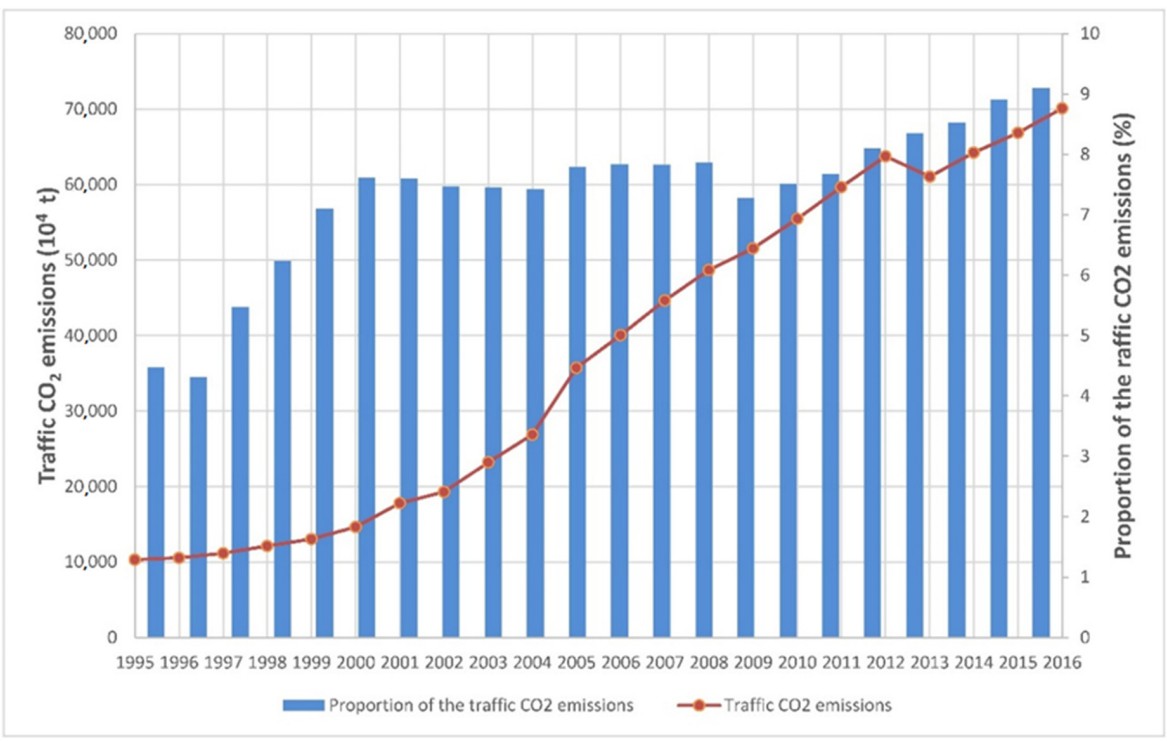

**Figure 1.** Total amount and contribution of transport $CO_2$ emissions in China during 1995 to 2016.

Both the amount and contributions of $CO_2$ emissions in the transport sector in China during the study period showed a 5-year periodical change, in accordance with the National Five-Year Plan. During the periods known as the Ninth Five-Year Plan (1995–2000) and the Tenth Five-Year Plan (2000–2005), the government focused mainly on economic development, which demands a great deal of transportation, and paid little attention to reducing $CO_2$ emissions. It should be noted that 2005 was a turning point. From 2005 to 2015, the period of the Eleventh Five-Year Plan and the Twelfth Five-Year Plan, measures and policies were formulated to optimize energy-use efficiency and to reduce greenhouse gas emissions. For example, the Outline of the National Medium- and Long-Term Program for Science and Technology Development (2006–2020) claimed that the transport industry should promote energy savings, resource conservation, and environmental protection and that breakthroughs in key technologies in the fields of resource saving and environmental protection should be widely applied.

Ten different fuel categories of national $CO_2$ emissions in the transport sector were evaluated, namely, raw coal, clean coal, coke, crude oil, gasoline kerosene, diesel oil, fuel oil, liquefied petroleum gas (LPG), and natural gas (shown in Figure 2). In 2016, the proportion of traffic $CO_2$ emissions of these different fuel categories was as follows: Diesel oil (48.38%) > gasoline (24.15%) > kerosene (14.52%) > fuel oil (7.55%) > raw coal (3.67%) > clean coal (0.93%) > natural gas (0.42%) > LPG (0.36%) > coke (0.01%) > crude oil (0.00%). Transport $CO_2$ emissions from diesel oil and gasoline always made up the greatest proportion during 1995 to 2016. During the period of 1995 to 2016, the proportion of transport $CO_2$ emissions from diesel oil, clean coal, LPG, and natural gas showed an increase. Though the proportions of transport $CO_2$ emissions from LPG and natural gas were still relatively low, the increasing trend indicated the vigorous development of new energy vehicles in China over the past two decades. The proportion of transport $CO_2$ emissions from gasoline, raw coal, coke, and crude oil decreased from 1995 to 2016.

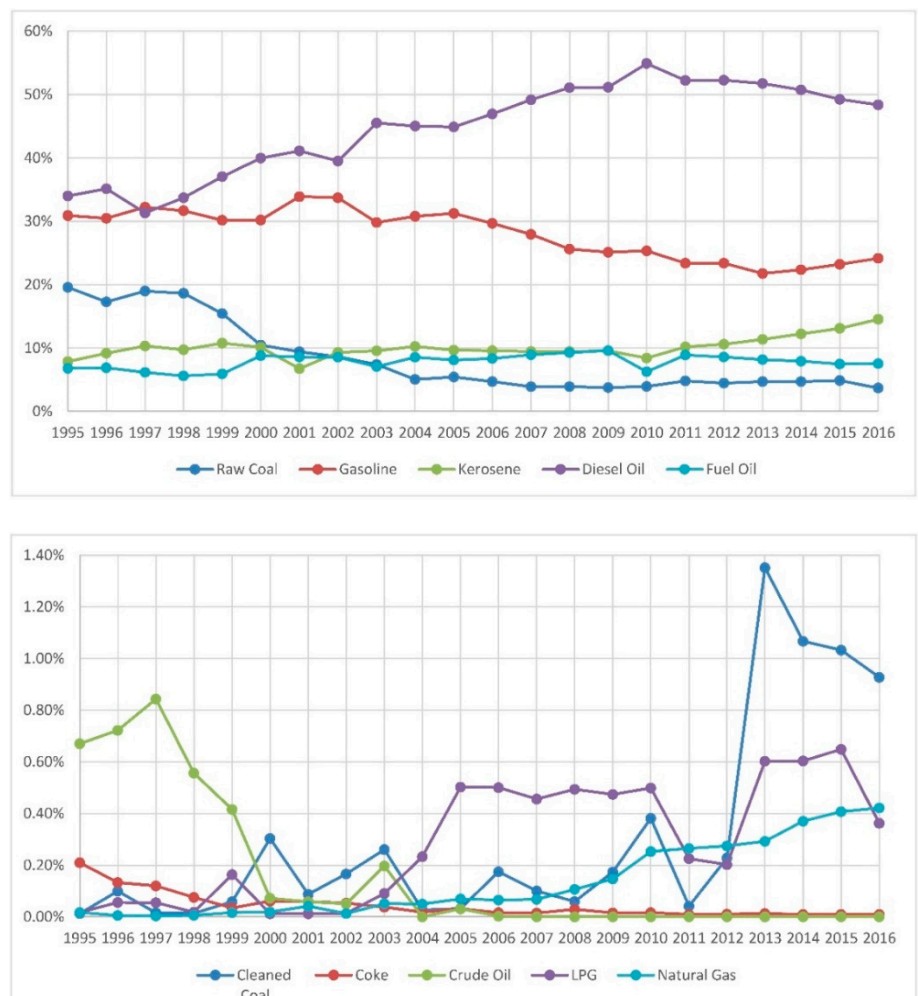

**Figure 2.** Temporal trends in the transport $CO_2$ emissions from 10 fuel categories in China.

3.1.2. Spatiotemporal Changes in Transport $CO_2$ Emissions

To uncover the current status and temporal characteristics of transport $CO_2$ emissions' spatial patterns in China, the provincial transport $CO_2$ emissions in 1995, 2000, 2005, 2010, 2015, and 2016 are shown in Figure 3.

Figure 3f depicts the traffic $CO_2$ emissions of 30 provinces in 2016. Most provinces have a relatively high level of $CO_2$ emissions in the transport sector above 10 Mt, except for Qinghai (3.24 Mt), Ningxia (3.35 Mt), Hainan (5.82 Mt), Tianjin (8.58 Mt), and Gansu (9.20 Mt). Rounding out the top 10 are Guangdong (68.24 Mt), Shanghai (48.34 Mt), Jiangsu (40.87 Mt), Shandong (39.83 Mt), Liaoning (38.86 Mt), Hubei (36.65 Mt), Zhejiang (30.90 Mt), Hunan (29.22 Mt), Sichuan (28.31 Mt), and Henan (25.64 Mt). Currently, the spatial pattern of transport $CO_2$ emissions in China shows an east-west differentiation. $CO_2$ emissions in the transport sector are higher in the east of China, especially in the eastern coastal region, and lower in the west, especially in the northwest regions, such as Qinghai, Ningxia, Gansu, and Shaanxi provinces. The current status of $CO_2$ emissions in the transport sector essentially parallels the current situation in regional economic development.

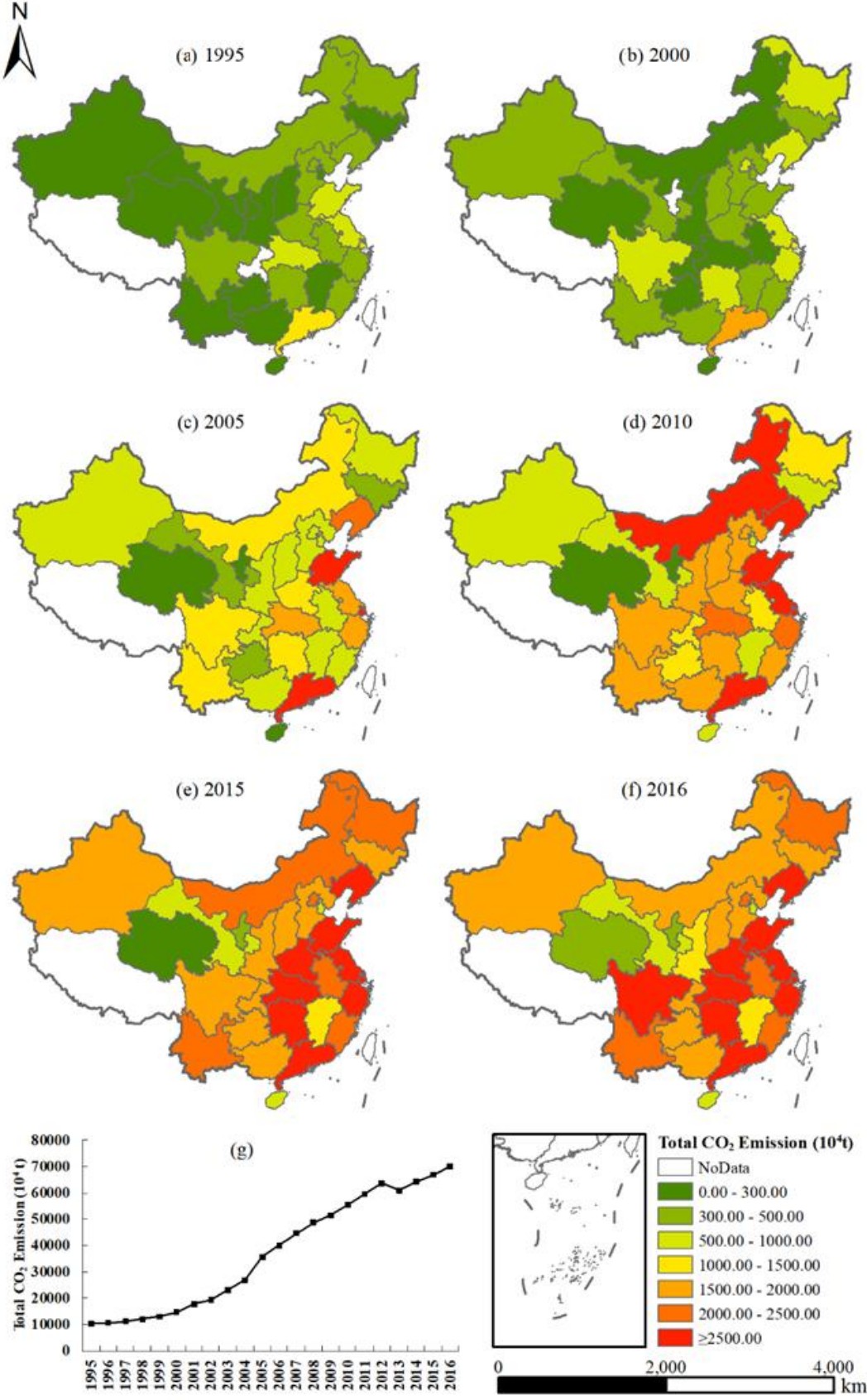

**Figure 3.** Transport CO$_2$ emissions across 30 provinces in China in 1995, 2000, 2005, 2010, 2015, and 2016 (**a–f**); and chart throughout the period of 1995 to 2016 (**g**).

The spatial pattern of $CO_2$ emissions in the transport sector in China changed greatly from 1995 to 2016 and can be divided into three phases: North-south differentiation (1995–2000), east-west differentiation (2000–2010), and differentiation in the northwest (2010–2016). In the first phase, transport $CO_2$ emissions were low in the north of China and high in the south, and they were especially high in the southern coastal region. The period of 2000 to 2005 was an important stage when the hotspots of transport $CO_2$ emissions moved northward. During 2000 to 2005, namely, the second phase, the north-south differentiation gradually disappeared, and transport $CO_2$ emissions became lower in the west of China and higher in the east, especially in the eastern coastal area. From 2005 to 2016, the hotspots of transport $CO_2$ emissions continuously expanded westward, especially to the central region and the southwest border area. In the third phase, transport $CO_2$ emissions in the northwest were lower than in other areas of China. Generally, the hotspots showed an expansion from the southeast coastal area to the eastern coastal area, and to the central area and the southwest border area.

From 1995 to 2016, $CO_2$ emissions in the transport sector of all provinces increased, except the Inner Mongolia Autonomous Region. Transport $CO_2$ emissions in Inner Mongolia decreased from 1995 to 2000, increased during 2000 to 2010, and decreased from 2010 to 2016. The Tenth and Eleventh Five-Year Plan for National Economic and Social Development of the Inner Mongolia Autonomous Region proposed the acceleration of a new type of industrialization during 2000 to 2010. The coal industry production was high during 2000 to 2010, especially during the Eleventh Five-Year Plan period, with an annual growth rate of 25% in raw coal productivity. After 2010, the government of Inner Mongolia paid more attention to the ecological environment, and the Twelfth Five-Year Plan for National Economic and Social Development of the Inner Mongolia Autonomous Region promoted green development and the construction of an environmentally friendly and resource-saving society.

### 3.1.3. Per Capita Transport $CO_2$ Emissions

The provincial transport $CO_2$ emissions per capita in 1995, 2000, 2005, 2010, 2015, and 2016 are shown in Figure 4. National transport $CO_2$ emissions per capita increased from 85.12 kg in 1995 to 507.00 kg in 2016. The annual average growth rate of national transport $CO_2$ emissions per capita was 8.87% during the period of 1995 to 2016, lower than the growth rate of the total amount of transport $CO_2$ emissions (9.56%). The variations in transport $CO_2$ emissions per capita were in accordance with the total amount of transport $CO_2$ emissions during the study period. From 1995 to 2000, transport $CO_2$ emissions per capita increased continuously at a rate of 5.82%. It increased rapidly from 2000 to 2005 at a growth rate of 19.31%. After 2005, the growth rate of transport $CO_2$ emissions per capita slowed down, to 8.66% during 2005 to 2010 and to 3.44% during 2010 to 2016. The analogous variation characteristics of the two variables indicated that transport $CO_2$ emissions per capita is an important factor influencing the total amount of transport $CO_2$ emissions during the study period. The growth rate of transport $CO_2$ emissions per capita was always lower than the total amount of transport $CO_2$ emissions. The difference between the growth rate of transport $CO_2$ emissions per capita and the total amount became less after 2000. This indicates that the influence of transport $CO_2$ emissions per capita on the total amount of transport $CO_2$ emissions has become more and more important recently.

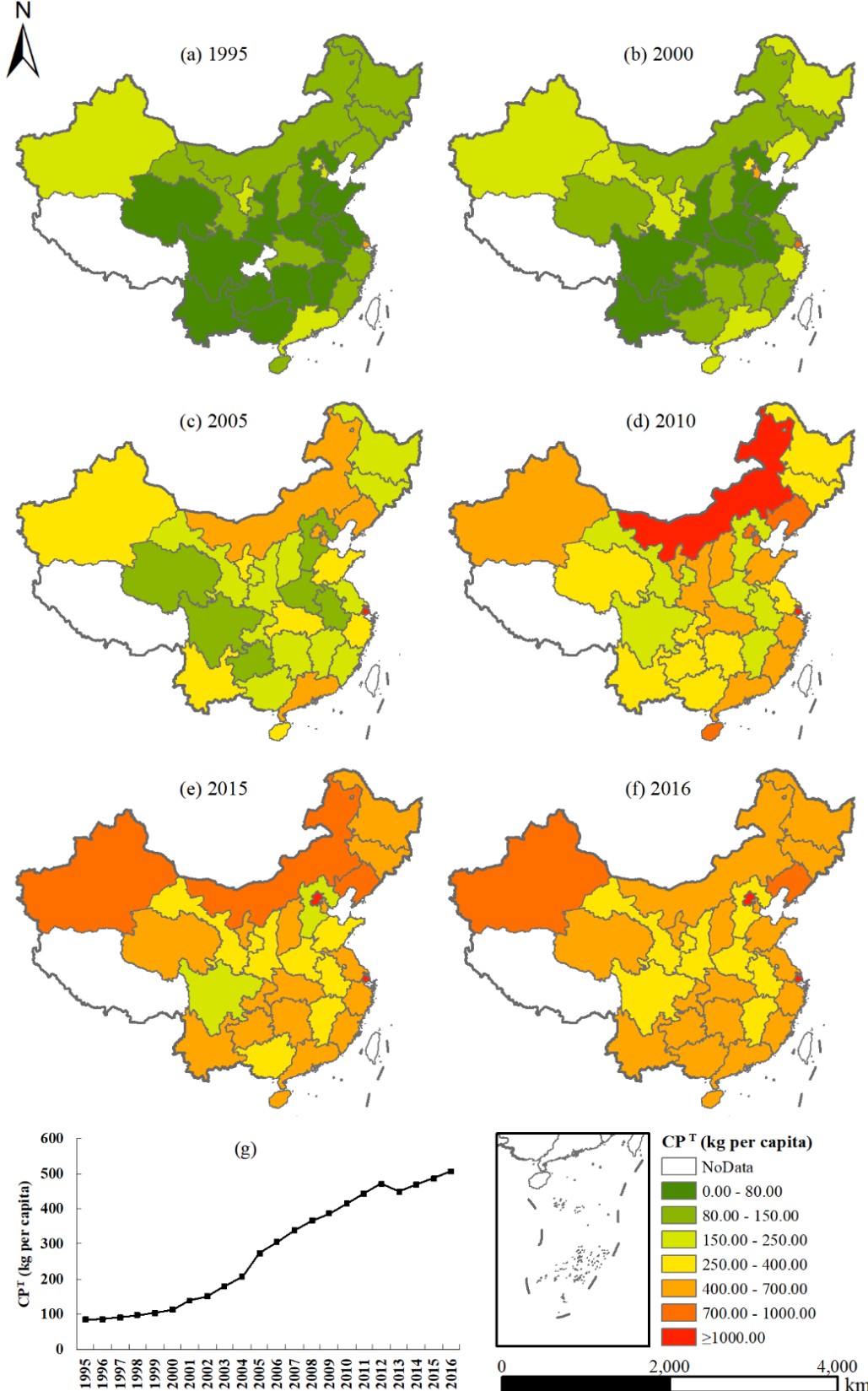

**Figure 4.** Transport $CO_2$ emissions per capita across 30 provinces in China in 1995, 2000, 2005, 2010, 2015, and 2016 (**a**–**f**); and chart throughout the period of 1995 to 2016 (**g**).

In 2016, the transport $CO_2$ emissions per capita in all provinces were above 300 kg, except Hebei (259.54 kg) and Henan (268.99 kg). Rounding out the top 10 were Shanghai (1997.60 kg), Beijing (1121.40 kg), Liaoning (887.67 kg), Xinjiang (763.88 kg), Inner Mongolia (667.05 kg), Hainan (635.03 kg), Hubei (622.7 kg), Guangdong (620.47 kg), Chongqing (608.94 kg), and Jilin (601.04 kg). In addition, the transport $CO_2$ emissions per capita in Shanghai were far higher than in other provinces, and transport $CO_2$ emissions per capita in Shanghai were nearly twice that of Beijing, the second largest transport $CO_2$ emissions province in China. Current provincial transport $CO_2$ emissions per capita showed a 'dumbbell' spatial pattern, higher in the east and west of China and lower in the central region. The high value of transport $CO_2$ emissions per capita in the west of China was mainly attributed to the primitive energy-saving and emission-reduction technology of automotive vehicles. Despite the fact that the economically developed eastern region of China has a relatively high level of carbon reduction technologies, the constant economic activity, improvements in living standards, and the increase in travel and transportation demand cause the eastern region of China to have a relatively high transport $CO_2$ emissions per capita.

Transport $CO_2$ emissions per capita of most provinces showed an increasing trend from 1995 to 2016, except for six provinces, including Inner Mongolia, Hainan, Tianjin, Shandong, Gansu, and Shaanxi. Generally, the spatial pattern of transport $CO_2$ emissions per capita in China from 1995 to 2016 can be divided into two phases: North-south differentiation (1995–2000) and the 'dumbbell' pattern (2000–2016). In the first phase, transport $CO_2$ emissions per capita were high in the north of China and low in the south, contrary to the total amount of traffic $CO_2$ emissions, for which the north was lower than the south. During the second phase (2000–2016), transport $CO_2$ emissions per capita in the southwest and eastern coastal regions increased noticeably and showed a 'dumbbell' spatial pattern. More attention should be paid to Shanghai, Guangdong, Liaoning, and Hubei. These four provinces are in the top 10 of transport $CO_2$ emissions, both in terms of the total amount and in terms of per capita. Attention should also be paid to Beijing, Xinjiang, Inner Mongolia, Hainan, Chongqing, and Jilin. These six provinces are included in the top 10 of transport $CO_2$ emissions per capita, although their total amount of transport $CO_2$ emissions are not included in the top 10 list.

$CO_2$ emissions in the transport sector are not only associated with socioeconomic development at the macroscopic level [36,37], but they are also related to lifestyle changes at the individual level [38]. Considering the differences in terms of the total amount and per capita in the temporal trends and spatial patterns of $CO_2$ emissions in the transport sector, together with the different meanings embodied in these two variables, the decoupling of $CO_2$ emissions in the transport sector should be analyzed in both aspects.

3.1.4. $CO_2$ Emissions Intensity in the Transport Sector (kg/$10^4$ yuan)

Figure 5 summarizes the national and provincial $CO_2$ emissions intensity in the transport sector in China from 1995 to 2016. The national transport $CO_2$ emissions intensity stayed at the level of 102 to 105 kg/$10^4$ yuan during 1995 to 2000. From 2000 to 2005, the national transport $CO_2$ emissions intensity fluctuated upward from 104.08 kg/$10^4$ yuan in 2000 to a peak value of 133.51 kg/$10^4$ yuan in 2005. The average annual growth rate was 5.11% during 2000 to 2005, much lower than the rate of national transport $CO_2$ emissions (19.54%) and national transport $CO_2$ emissions per capita (19.31%). The year, 2005, was a turning point for national transport $CO_2$ emissions intensity in China. After 2005, the intensity declined to 89.87 kg/$10^4$ yuan in 2016, even 22.57 kg/$10^4$ yuan lower than in 1995. Since 2005, a series of policies have been proposed by the government. In 2006, the Renewable Energy Law of People's Republic of China was enacted to promote the development and utilization of renewable energy. In 2007, China's National Climate Change Program was issued, which proposed the elimination of old passenger cars and vessels with high fuel consumption. In 2009, the Measures for Testing and Supervision of Fuel Consumption of Vehicles and Notice of Demonstration and Promotion of the Energy Saving and New Energy Vehicles were released. These two measures promoted the enhancement of energy-saving management of vehicles and encouraged energy saving and new-energy vehicles by financial means. The Twelfth Five-Year Plan for Energy Saving and Emission Reduction of Highway and Waterway Transportation, issued in 2010, proposed establishing a

low-carbon transportation system. In 2013, the Instructions of Accelerating the Development of Green, Circular and Low-Carbon Transportation promoted the construction of a low-carbon transportation system by introducing cyclical development into the system. It should be noted that the government played an important role in low-carbon transportation development and that more such positive actions should be taken to achieve further improvement.

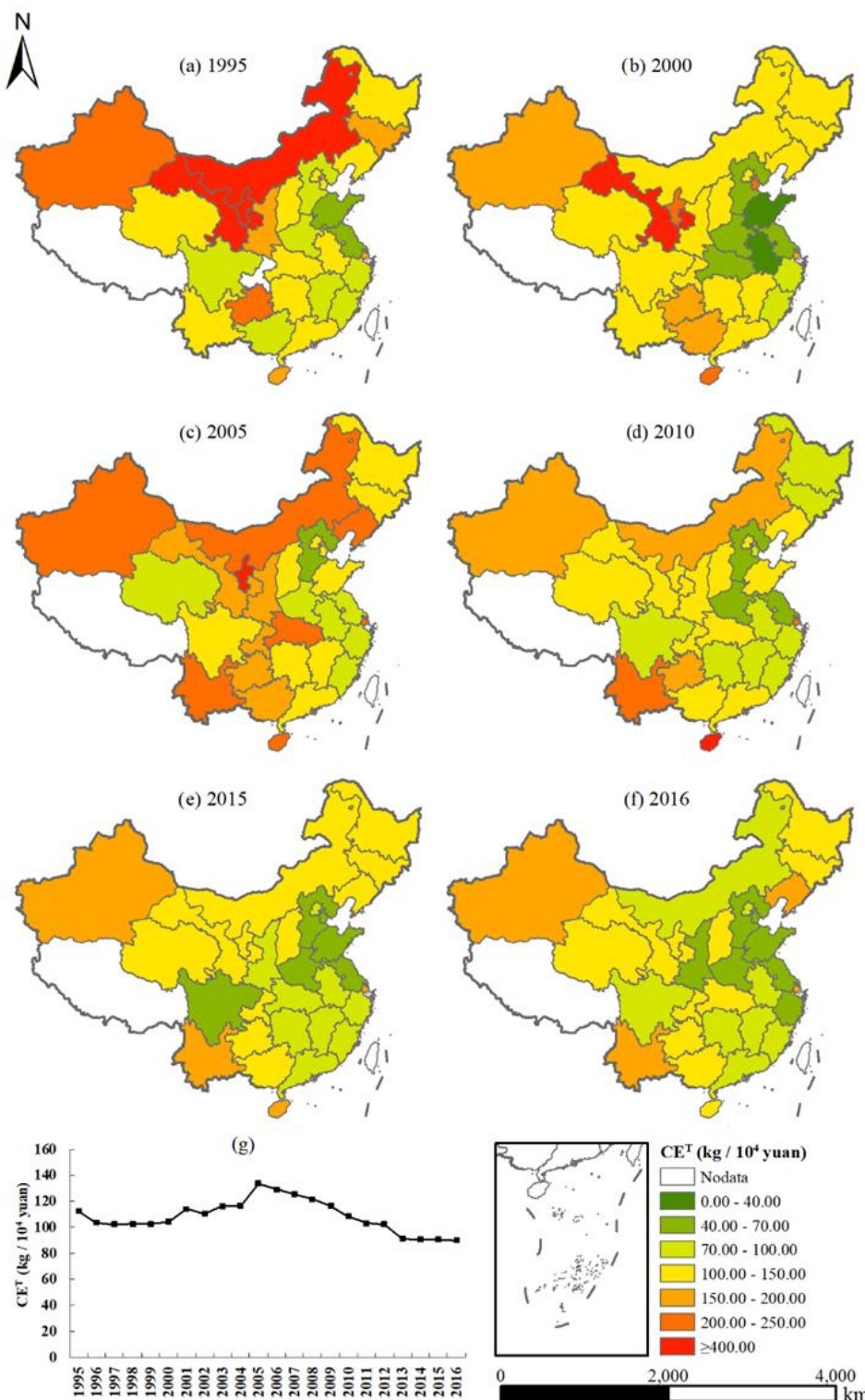

**Figure 5.** Transport CO$_2$ intensity in China in 1995, 2000, 2005, 2010, 2015, and 2016 (**a–f**); and chart throughout the period of 1995 to 2016 (**g**).

Generally, provincial transport $CO_2$ emissions intensity showed an east-west differentiation during 1995 to 2016, and transport $CO_2$ emissions intensity was higher in the west of China and lower in the east, which showed an inverse relationship to the spatial pattern of the economic and technological levels. In 2016, the highest figure of 189.83 kg/$10^4$ yuan was recorded in Xinjiang, and the lowest figure of 47.97 kg/$10^4$ yuan was found in Tianjin. The transport $CO_2$ emissions intensity of four provinces, including those of Zhejiang, Henan, Hebei, and Jiangsu, were below 100 kg/$10^4$ yuan throughout the period of 1995 to 2016.

### 3.2. Analysis of the Decoupling between Transport $CO_2$ Emissions and Economic Growth

Given that most economic, social, and environmental plans in China are five-year plans, and given the five-year periodic trends in transport $CO_2$ emissions intensity and transport $CO_2$ emissions from both the total and individual perspective, the decoupling of $CO_2$ emissions in the transport sector from economic growth was measured over four periods: 1995–2000, 2000–2005, 2005–2010, and 2010–2015. Figure 6 shows the provincial decoupling states for the above four periods in terms of total amount. No observations fall in category IIb, which represents the worst case situation and denotes absolute strong coupling. As a result of economic growth over the last two decades in China, all observations fall in four categories as follows: Category Ia denotes strong relative coupling, Ib denotes weak relative decoupling, IIIa denotes weak absolute decoupling, and IIIb denotes strong absolute decoupling. As a reminder, IIIb refers to the most desirable situation with good local practices.

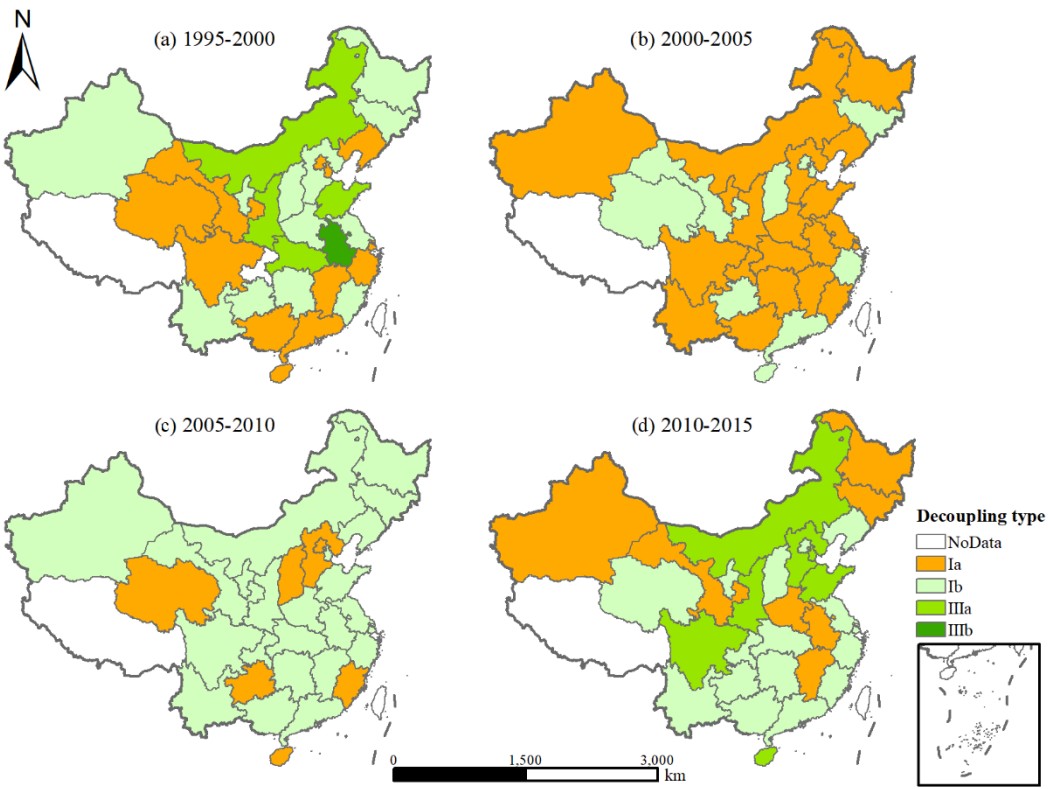

**Figure 6.** Decoupling state of transport $CO_2$ emissions from economic growth. Ia denotes the strong relative coupling, Ib denotes the weak relative decoupling, IIIa denotes the weak absolute decoupling, and IIIb denotes the strong absolute decoupling. IIIb refers to the most desirable situation.

During 1995 to 2000, strong absolute decoupling appeared in Anhui province, weak absolute decoupling appeared in four provinces (Inner Mongolia, Shandong, Shaanxi, and Hubei), and weak relative decoupling appeared in 12 other provinces. These 12 provinces experienced an increase in both GDP and transport $CO_2$ emissions, and the GDP growth rate was lower than that for transport

$CO_2$ emissions. The Bohai Rim region, Yangtze River Delta, Pearl River Delta, and the northwest region experienced more rapid growth in transport $CO_2$ emissions than in their economy. It is notable that the decoupling worsened in most provinces from 2000 to 2005, during the Tenth Five-Year Plan period. Strong relative coupling occurred in 20 provinces in China, and only 10 provinces (Jilin, Beijing, Tianjin, Shanxi, Gansu, Qinghai, Zhejiang, Guangdong, Hainan, and Guizhou) showed weak relative decoupling. During 2005 to 2010, an apparent improvement took place in comparison to 2000 to 2005. Weak relative decoupling took place in more than 70% of provinces in China, and only seven provinces (Beijing, Hebei, Shanxi, Qinghai, Guizhou, Hainan, and Fujian) showed strong relative coupling. Although the decoupling was only in the weak relative decoupling category, a turn for the better can definitely be detected. During 2010 to 2015, seven provinces (Heilongjiang, Jilin, Xinjiang, Gansu, Henan, Anhui, and Jiangxi) did not achieve decoupling, and weak absolute decoupling appeared in seven provinces (Inner Mongolia, Hebei, Tianjin, Shandong, Shaanxi, Sichuan, and Hainan). There were 16 provinces that showed weak relative decoupling, and decoupling took place in nearly 80% of provinces in China during 2010 to 2015.

The decarbonisation state in the transport sector showed periodic changes. During the Tenth Five-Year Plan period, coupling included nearly 70% of provinces in China, and only 10 provinces experienced weak relative decoupling. During the Eleventh and Twelfth Five-Year Plan period, significant progress was made in the decarbonisation of the transport sector. The characteristics of the periods are indicative of the influence of national policies on $CO_2$ reductions in the transport sector. However, the decoupling achieved was almost all relative decoupling, and no strong absolute decoupling took place. Therefore, transport $CO_2$ reduction policies are not enough to achieve the most desirable decoupling state. Improvements in technology, including the development of renewable energy, utilization of new energy vehicles, eco-driving behaviors, and an intelligent transportation system, are urgently needed.

Table 3 shows the decoupling index in terms of the total amount (D) and per capita (DP) of the 30 provinces during the four periods. The D and DP indices of five provinces (Inner Mongolia, Anhui, Shandong, Hubei, and Shaanxi) during 1995 to 2000 and seven provinces (Tianjin, Hebei, Inner Mongolia, Shandong, Hainan, Sichuan, and Shaanxi) during 2010 to 2015 were below zero. This shows that these provinces achieved an absolute decoupling of transport $CO_2$ emissions from economic growth in terms of both the total amount and per capita during the same period. The practices of these five provinces during 1995 to 2000 and these seven provinces during 2010 to 2015 are worth studying. Although Hebei and Jilin did not achieve absolute decoupling in terms of the total amount during 1995 to 2000, an absolute decoupling of transport $CO_2$ emissions from economic growth in terms of per capita was achieved in these provinces during 1995 to 2000. The practices for the reduction of transport $CO_2$ emissions from the individual perspective in Hebei and Jilin from 1995 to 2000 should also be paid attention to. In addition, the D indices of five of the provinces (Tianjin, Liaoning, Guangdong, Guangxi, and Sichuan) kept declining during 1995 to 2015, and the decoupling state of these five provinces has improved over the past two decades. In particular, Tianjin not only showed continuous progress in the decoupling of transport $CO_2$ emissions from economic growth, but it also achieved absolute decoupling from 2010 to 2015, the closest Five-Year Plan period. In 2004, the Measures for Supervision and Test Energy Saving Management in Tianjin were proposed, and in 2010, Tianjin's Program to Address Climate Change was carried out. It emphasized energy savings and emission reductions that have no negative effects on the environment when the economy and society develop rapidly. The year after, Tianjin suggested that it will explore a path for the creation of a compact city, prompt the construction of a green and low-carbon transportation system, enhance public transportation, implement energy standards above national standards, promote energy-saving and new-energy vehicles, develop energy efficiency assessment and energy-saving indices management, and develop the low-carbon lifestyle of its residents. In general, the policies in Tianjin covered various aspects, from the city-level to the enterprise-level, and even down to the individual-level.

**Table 3.** Comparison of D and DP across 30 provinces in China.

| Region | D (Decoupling Index) | | | | DP (Decoupling Index in the Sense of Per Capita) | | | |
|---|---|---|---|---|---|---|---|---|
| | 1995–2000 | 2000–2005 | 2005–2010 | 2010–2015 | 1995–2000 | 2000–2005 | 2005–2010 | 2010–2015 |
| Beijing | 1.19 | 0.55 | 1.12 | 0.60 | 1.23 | 0.51 | 1.18 | 0.46 |
| Tianjin | 2.46 | 0.38 | 0.27 | −0.13 | 2.63 | 0.35 | 0.07 | −0.70 |
| Hebei | 0.03 | 1.22 | 1.28 | −0.09 | −0.05 | 1.22 | 1.30 | −0.23 |
| Shanxi | 0.72 | 0.93 | 1.12 | 0.76 | 0.66 | 0.93 | 1.13 | 0.73 |
| Inner Mongolia | −0.44 | 2.44 | 0.64 | −0.24 | −0.55 | 2.45 | 0.63 | −0.30 |
| Liaoning | 1.95 | 2.58 | 0.32 | 0.54 | 2.02 | 2.57 | 0.29 | 0.54 |
| Jilin | 0.05 | 0.73 | 0.88 | 1.90 | −0.07 | 0.73 | 0.88 | 1.91 |
| Heilongjiang | 0.81 | 1.03 | 0.26 | 4.04 | 0.81 | 1.03 | 0.26 | 3.98 |
| Shanghai | 1.46 | 1.57 | 0.65 | 0.28 | 1.63 | 1.67 | 0.46 | 0.13 |
| Jiangsu | 0.68 | 1.61 | 0.59 | 0.78 | 0.65 | 1.62 | 0.57 | 0.78 |
| Zhejiang | 1.13 | 0.86 | 0.66 | 0.82 | 1.15 | 0.86 | 0.62 | 0.81 |
| Anhui | −1.48 | 5.33 | 0.65 | 1.78 | −1.45 | 5.48 | 0.66 | 1.83 |
| Fujian | 0.48 | 1.55 | 1.05 | 0.58 | 0.42 | 1.58 | 1.05 | 0.55 |
| Jiangxi | 2.57 | 1.02 | 0.35 | 1.06 | 2.64 | 1.02 | 0.33 | 1.07 |
| Shandong | −0.47 | 8.09 | 0.56 | −0.63 | −0.59 | 8.22 | 0.54 | −0.74 |
| Henan | 0.06 | 1.51 | 0.64 | 1.22 | 0.03 | 1.52 | 0.64 | 1.23 |
| Hubei | −0.77 | 14.83 | 0.23 | 0.33 | −0.90 | 13.30 | 0.23 | 0.30 |
| Hunan | 0.97 | 1.61 | 0.41 | 0.90 | 0.97 | 1.59 | 0.39 | 0.89 |
| Guangdong | 1.26 | 0.88 | 0.51 | 0.31 | 1.49 | 0.87 | 0.41 | 0.23 |
| Guangxi | 6.82 | 1.19 | 0.62 | 0.34 | 6.43 | 1.20 | 0.63 | 0.28 |
| Hainan | 2.96 | 0.82 | 1.31 | −0.07 | 3.75 | 0.80 | 1.33 | −0.17 |
| Chongqing | - | 2.18 | 0.54 | 0.69 | - | 2.06 | 0.53 | 0.67 |
| Sichuan | 10.99 | 1.17 | 0.73 | −0.04 | 2.25 | 1.16 | 0.74 | −0.08 |
| Guizhou | 0.81 | 0.72 | 1.29 | 0.49 | 0.80 | 0.70 | 1.27 | 0.48 |
| Yunnan | 0.99 | 3.68 | 0.77 | 0.29 | 0.99 | 3.85 | 0.76 | 0.26 |
| Tibet | - | - | - | - | - | - | - | - |
| Shaanxi | −0.13 | 1.98 | 0.82 | −0.20 | −0.19 | 2.00 | 0.82 | −0.24 |
| Gansu | 1.61 | 0.00 | 0.26 | 1.30 | 1.67 | 0.01 | 0.26 | 1.31 |
| Qinghai | 1.88 | 0.26 | 2.02 | 0.43 | 2.06 | 0.21 | 2.06 | 0.38 |
| Ningxia | 0.30 | 1.11 | 0.20 | 0.21 | 0.09 | 1.13 | 0.17 | 0.11 |
| Xinjiang | 0.45 | 1.41 | 0.32 | 1.59 | 0.21 | 1.44 | 0.24 | 1.71 |

The D index of 11, 13, 8, and 6 provinces was lower than the DP index during 1995 to 2000, 2000 to 2005, 2005 to 2010, and 2010 to 2015, respectively. The D index of 15, 11, 15, and 22 provinces was higher than the DP index during the above four periods. This finding indicates that the decoupling of transport $CO_2$ emissions from economic growth from the perspective of the total amount was better than from the individual perspective when the D index was lower than the DP index. The year, 2005, was a turning point. Before 2005, the proportion of provinces with a higher D or a higher DP index was similar. After 2005, the proportion of the provinces with a higher D was much higher than DP. The important achievements in decoupling after 2005 were more due to changes at the individual level. Policies that encouraged public transport and eco-driving behaviors had a definite effect on the decoupling of transport $CO_2$ emissions from economic growth. To further achieve absolute decoupling from both the individual perspective and the total amount, these measures should be paid more attention to: Maintenance of long-term eco-driving behaviors, promotion and application of energy-saving technologies and renewable energy, construction of an intelligent and efficient transportation system, and flexible working schedules to alleviate the overloading of road transportation in metropolitan areas.

## 4. Conclusions

To show the decoupling of $CO_2$ emissions in the transport sector from economic growth and to identify good local practices for reducing transport $CO_2$ emissions in China, this paper estimated $CO_2$ emissions in the transport sector of 30 provinces from 1995 to 2016 and examined the provincial decoupling of $CO_2$ emissions in the transport sector from economic growth. In addition, this paper developed a decoupling indicator in terms of per capita in the framework of the Tapio decoupling model

to better understand the relationship between $CO_2$ emissions and economic growth for developing countries and under-developed regions. The key findings can be summarized as follows: (1) The 5-year periodical change, which was also observed in the power industry in China [39], and the turning point in 2005 in $CO_2$ emissions in the transport sector indicated that national carbon reduction policies played a significant role in partially achieving a low-carbon transition in the transport sector during 1995 to 2016. (2) The inverse spatial patterns of transport $CO_2$ emissions in terms of the total amount and per capita—'higher in east lower in west'—and the transport $CO_2$ intensity—'lower in east higher in west'—indicate that traffic energy-saving technologies improved with economic growth and had an influence on $CO_2$ reduction in the transport sector, and depending only on improvements in low-carbon technologies was not enough to achieve absolute $CO_2$ reduction in the transport sector. (3) Although the decoupling states improved a lot, great challenges still exist for achieving absolute decoupling, especially strong absolute decoupling. Decoupling was achieved in nearly 80% of provinces in China during 2010 to 2015. However, most of the decoupling was relative decoupling, and absolute decoupling only took place in seven provinces (Inner Mongolia, Hebei, Tianjin, Shandong, Shaanxi, Sichuan, and Hainan). (4) A comparison of the indices, *D* and *DP*, revealed the significant influence of transport $CO_2$ reduction policies from the individual perspective. More policies to achieve decoupling in the transport sector from economic growth from the perspective of the total amount are urgently needed. (5) Tianjin has good local practices for achieving decoupling in the transport sector. The integration of multi-scale traffic $CO_2$ reduction policies in Tianjin should also be promoted in Beijing and Hebei as part of the national strategy of the Beijing-Tianjin-Hebei Cooperative Development in advance and then promoted in other regions.

Interregional $CO_2$ reduction cooperation in the transport sector should be promoted. To further achieve the strong absolute decoupling state, especially in terms of total amount, the transfer of $CO_2$ reduction technology in the transport sector from developed areas to underdeveloped areas should be facilitated. The Beijing-Tianjin-Hebei region should be taken as the center and the north and northeast provinces should be promoted, including Shandong, Shanxi, Inner Mongolia, Liaoning, Jilin, and Heilongjiang. The Yangtze River Delta region should be taken as the center and central China promoted, including Anhui, Hubei, Hunan, and Jiangxi. Guangdong should be taken as the center and Guangxi, Gujian, and Hainan promoted. In addition, one-to-one traffic $CO_2$ reduction technical assistance relationships on the basis of poverty alleviation assistance relationships need to be created. For example, Tianjin exports $CO_2$ reduction technology by transporting to Gansu, Yunnan and Guizhou import technology from Shanghai, etc. However, transport $CO_2$ emissions in terms of both the total amount and individual are great in developed areas, although the transport $CO_2$ intensity is lower. This phenomenon is due to the large demand of travel derived from economic activities. Therefore, flexible working schedules and carpooling may become effective transportation demand management measures for alleviating transportation overload and traffic jams in developed areas. In February 2019, Price Waterhouse Coopers Consulting in China implemented a flexible working schedule. Carpooling is becoming fashionable in metropolitan areas, such as Beijing, Shanghai, and Guangzhou. Improving the safety of carpooling platforms to encourage the sharing of travel resources in both inner-city and inter-city transportation is also an important path for reducing transport $CO_2$ emissions.

Due to the limitation of energy consumption statistics in the transport sector in China, a certain uncertainty exists in this study. Although the uncertainty exists, it did not significantly influence the regularities and results since the main aim of this study was to explore the spatiotemporal pattern and uncover its decoupling status from economic growth. With the continuous improvements in the energy consumption statistics in China, more accurate results of the transport $CO_2$ emissions can be obtained in the future. In addition, the identification of the driving forces of the decoupling of transport $CO_2$ emissions from economic growth is of great significance to achieve a strong absolute decoupling state. Therefore, identifying influencing factors of decoupling transport $CO_2$ emissions from economic growth and quantifying theirs effects on the decoupling status will be an important topic in the next step.

**Author Contributions:** Research framework: J.Z., Y.H. and S.D.; methodology: J.Z., Y.H. and Y.L.; data collection: Y.H. and J.Z.; results analysis: J.Z., Y.L. and Y.H.; writing: J.Z., Y.H. and Y.L.; revising: J.Z., S.D., Y.H., Y.L.

**Funding:** This research was funded by National Natural Science Foundation of China, Grant No. 41771182, Science & Technology Basic Resources Investigation Program of China, Grant No. 2017FY101300, 2017FY101303, and the National Social Science Fund of China, Grant No. 17VDL016.

**Conflicts of Interest:** The authors have no conflicts of interest to declare.

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
