# Peer review of "The Spatiotemporal Pattern of Decoupling Transport CO2 Emissions from Economic Growth across 30 Provinces in China"

_sustainability, doi:10.3390/su11092564_

Round 1

Reviewer 1 Report

Thank you for the opportunity to review this interesting paper. I have few remarks concerning the research:

* First of all authors state that they developed an extended decoupling indicator in terms of per capita in order to determine the spatial patterns and temporal trends of the decoupling of transport CO2 emissions from economic growth. The presented methodology does not discuss clear enough what extensions authors made to already developed models and how they allow to determine the spatial patterns and temporal trends of the decoupling of transport CO2 emissions from economic growth. Does it mean that previously developed methods do not allow doing that? My suggestion is to clarify by emphasizing where are the essential differences between authors’ proposed and previous models, because now it seems that the only difference is measuring variables not in absolute terms but in per capita terms.

*I would also encourage authors to find a more compact way to present and discuss data and findings. Now presentation of the data and discussion is divided in to five-year periods and it is hard to follow the trend of changes.

*In my opinion, discussed policy implications are too broad and do not derived directly from the empirical findings. It is clear without any research that " maintenance of long-term eco-driving behaviors, promotion and application of energy-saving technologies and renewable energy, construction of an intelligent and efficient transportation system, and flexible working schedules to alleviate the overload on road transportation in metropolitan areas" would reduce CO2 emissions from transportation. My suggestion is to concentrate on policy implications that could be directly grounded by empirical findings of the research.

Author Response

Response to the reviewers’ comments:

Reviewer #1

Thank you for the opportunity to review this interesting paper. I have few remarks concerning the research.

Thank you very much for your guidance and concern of our manuscript entitled “The spatiotemporal pattern of decoupling transport CO2 emissions from economic growth across 30 provinces in China” (Manuscript ID: sustainability-491083). Your comments are greatly helpful for revising and improving our manuscript, as well as have important guiding significance to our research. According to your guidance, we have made detailed revisions in our revised manuscript. We hope that the revised manuscript can meet your approval. Revised portion are marked in the manuscript. The detail (point-by-point) list of the response is as follows:

ü  First of all authors state that they developed an extended decoupling indicator in terms of per capita in order to determine the spatial patterns and temporal trends of the decoupling of transport CO2 emissions from economic growth. The presented methodology does not discuss clear enough what extensions authors made to already developed models and how they allow to determine the spatial patterns and temporal trends of the decoupling of transport CO2 emissions from economic growth. Does it mean that previously developed methods do not allow doing that? My suggestion is to clarify by emphasizing where are the essential differences between authors’ proposed and previous models, because now it seems that the only difference is measuring variables not in absolute terms but in per capita terms.

Thank you very much for your precious suggestions on discussing the improvement of the method we developed in our manuscript more clearly. We developed a decoupling indicator in terms of per capita under the framework of the developed Tapio’s decoupling model to better capture and understand the relationship between transport CO2 emissions and economic growth from the aspect of per capita and provide scientific support to formulate CO2 emission mitigation policies related to the individual characteristics and behaviours. In addition, this study explores the decoupling of transport CO2 emissions from economic growth covering almost all provinces in China and more than two decades period. This allows the spatial pattern and temporal trends analysis. According to your suggestions, we have separate the Section ‘An extended Tapio decoupling model’ into two sections as ‘Tapio decoupling model’ and ‘Decoupling indicator in terms of per capita and decoupling typology’ to highlight the improvement we make in the method.

The methodology section is revised as follows:

2.3. Tapio decoupling model

The decoupling model proposed by Tapio [14] is widely used to analyse the relationship between economic growth and negative environmental externalities [31,32], especially CO2 emissions in the transport sector [33]. The decoupling index from year T-j to year T can be calculated as follows:

(2)

where  denotes the decoupling index from year T-j to year T; denotes the CO2 emissions change percentage from year T-j to year T; and  denotes the economic growth change percentage from year T-j to year T.  and  can be calculated as follows:

(3)

The relationship between GDP and CO2 emissions in the transport sector at a given time can be calculated as follows:

                               (5)

where denotes the transport CO2 emissions intensity per unit GDP.

2.4. Decoupling indicator in terms of per capita and decoupling typology

To better understand and reduce the transport CO2 emissions in developing countries while they strive for economic development, the decoupling index in terms of per capita is proposed as follows:

where   denotes the decoupling index in terms of per capita from year T-j to year T;  denotes the transport CO2 emissions per capita change percentage from year T-j to year T; and denotes the GDP per capita change percentage from year T-j to year T.

The degrees of decoupling between transport CO2 emissions and GDP are divided into 8 types [6], and the typology is shown in Table 2.”

ü  I would also encourage authors to find a more compact way to present and discuss data and findings. Now presentation of the data and discussion is divided in to five-year periods and it is hard to follow the trend of changes.

Thank you so much for your suggestions on the interval of data analysis. Before the decoupling analysis, the investigation of provincial transport CO2 emissions in terms of total amount, per capita and intensity is conducted. The significant 5-year periodic trend is observed during the study period and it’s coincided with the national and provincial five-year plan. Considering the characteristics of transport CO2 emission and the time period of major policies’ formulation comprehensively, the 5-year interval decoupling results analysis and discussion is appropriate and acceptable.

In addition, Tapio presented decoupling analysis results at 10-year interval in the research paper that he established the DT indicator denoting the emission-to-economic activity elasticity. Several recent research related to the decoupling the CO2 emissions of various sectors from economic growth in China also discussed data and findings at 5-year interval, such as ‘Yang, L., Yang, Y., Zhang, X., Tang, K. Whether China’s industrial sectors make efforts to reduce CO2 emissions from production? – A decomposed decoupling analysis. Energy. 2018, 160, 796-809.’ and ‘Wu., Y., Tam V.W.Y., Shuai, C., Shen, L., Zhang, Y., Liao, S. Decoupling China’s economic growth from carbon emissions: Empirical studies from 30 Chinese provinces (2001-2015). Sci. Total Environ. 2019, 666, 576-588.’.

ü  In my opinion, discussed policy implications are too broad and do not derived directly from the empirical findings. It is clear without any research that " maintenance of long-term eco-driving behaviors, promotion and application of energy-saving technologies and renewable energy, construction of an intelligent and efficient transportation system, and flexible working schedules to alleviate the overload on road transportation in metropolitan areas" would reduce CO2 emissions from transportation. My suggestion is to concentrate on policy implications that could be directly grounded by empirical findings of the research.

Thank you for your precious comments on the policy discussion. According to your suggestion, we have rewritten the content relate to the policy. The content about the interregional CO2 reduction cooperation in the transport on the basis of the spatiotemporal pattern of transport CO2 emissions and its decoupling state from economic growth is proposed. The broad policy implications about the long-term eco-driving behaviours, energy saving technologies and efficient transportation system have been canceled. In addition, we also add the carpooling as an important way to reduce transport CO2 emissions in developed area in China.

The revised Conclusion section is as follows:

“Conclusions

To show the decoupling of CO2 emissions in the transport sector from economic growth and to identify good local practices for reducing transport CO2 emissions in China, this paper estimated CO2 emissions in the transport sector of 30 provinces from 1995 to 2016 and examined the provincial decoupling of CO2 emissions in the transport sector from economic growth. In addition, this paper developed a decoupling indicator in terms of per capita in the framework of Tapio decoupling model to better understand the relationship between CO2 emissions and economic growth for developing countries and under-developed regions. The key findings can be summarised as follows: (1) The 5-year periodical change and the turning point in 2005 in CO2 emissions in the transport sector indicated that national carbon reduction policies played a significant role in partially achieving a low-carbon transition in the transport sector during 1995-2016. (2) The inverse spatial patterns of transport CO2 emissions in terms of total amount and per capita, ‘higher in east lower in west’, and the transport CO2 intensity, ‘lower in east higher in west’, indicate that traffic energy-saving technologies improved with economic growth and had an influence on CO2 reduction in the transport sector, and depending only on improvements in low-carbon technologies was not enough to achieve absolute CO2 reduction in the transport sector. (3) Although the decoupling states improved a lot, great challenges still exist for achieving absolute decoupling, especially strong absolute decoupling. Decoupling was achieved in nearly 80% of provinces in China during 2010-2015. However, most of the decoupling was relative decoupling, and absolute decoupling only took place in seven provinces (Inner Mongolia, Hebei, Tianjin, Shandong, Shaanxi, Sichuan and Hainan). (4) A comparison of indices D and DP revealed the significant influence of transport CO2 reduction policies from the individual perspective. More policies to achieve decoupling in the transport sector from economic growth from the perspective of total amount are urgently needed. (5) Tianjin has good local practices for achieving decoupling in the transport sector. The integration of multi-scale traffic CO2 reduction policies in Tianjin should also be promoted in Beijing and Hebei as part of the national strategy of the Beijing-Tianjin-Hebei Cooperative Development in advance and then promoted in other regions.

Interregional CO2 reduction cooperation in the transport sector should be promoted. To further achieve the strong absolute decoupling state, especially in terms of total amount, facilitate CO2 reduction technology in the transport sector transfer from the developed area to underdeveloped area. Take Beijing-Tianjin-Hebei region as center and promote the north and northeast provinces, including Shandong, Shanxi, Inner Mongolia, Liaoning, Jilin and Heilongjiang. Take Yangtze River Delta region as center and promote central China, including Anhui, Hubei, Hunan and Jiangxi. Take Guangdong as center and promote Guangxi, Gujian and Hainan. In addition, constructing one-to-one traffic CO2 reduction technical assistance relationship on the basis of poverty alleviation assistance relationship. For example, Tianjin exports CO2 reduction technology in the transport to Gansu, Yunnan and Guizhou imports technology from Shanghai, etc. However, the transport CO2 emissions in terms of both total amount and individual are great in developed area, although the transport CO2 intensity are lower. This phenomenon is due to large demand of travel derived from the economic activities. Therefore, the flexible working schedules and carpooling may become effective transportation demand management measure for alleviating transportation overload and traffic jams in developed area. In February 2019, Price Waterhouse Coopers Consulting in China implemented a flexible working schedule. Carpooling is becoming fashionable in metropolitan areas, such as Beijing, Shanghai and Guangzhou. Improving the safety of carpooling platforms to encourage the sharing of travel resources in both inner-city and inter-city transportation is also an important path for reducing transport CO2 emissions.

The above is our modification. Thank you again for your warm hearted guidance. We hope that our revised manuscript can meet your approval. We have tried our best to improve our manuscript and made some corrections in the manuscript. These corrections will not influence the main content of this paper. We appreciate for Editors/Reviewers’ warm work earnestly, and hope that the correction will meet with approval. If you have any question, please don’t hesitate to contact us. Thank you very much!

Sincerely,

Suocheng Dong, Yu Li

Reviewer 2 Report

The authors should clarify what kind of transportation is included in this study? are the emissions from railway and ship included? It is important for identifying driving forces. 

Figure 2 is confusing.

Author Response

ü  The authors should clarify what kind of transportation is included in this study? are the emissions from railway and ship included?

Thank you very much for your precious suggestions on clarifying the kind of transportation in this study. The energy consumption data in the transportation sector of 30 provinces are collected from China Energy Statistical Yearbook (1996-2017). According to the Classification of National Economic Industries (GB/T 4754-2017), the transportation, warehousing and postal service includes railway transportation industry, road transportation industry, ship transportation industry, air transportation industry, pipeline transportation industry, multimodal transport and transport agent industry, handling and warehousing industry and postal industry. The main aim of this manuscript is to explore the spatiotemporal pattern of the CO2 emissions in the transport sector and uncover its decoupling characteristics from economic growth. We acknowledge that limited to the deficiency of the statistics of energy consumption in the transport sector, a certain degree of uncertainty exists in this study, but it won’t significantly influence results and regularities in this manuscript. The clarification content is supplied in the data description and the discussion on the uncertainty is added in the conclusion section.

The supplement content in the data description section is as follows:

2.1. Data description

The provincial energy consumption data used to calculate the CO2 emissions in the transport sector in China from 1995 to 2016 are collected from the China Energy Statistical Yearbook [28]. Ten types of fuels are considered in this study: raw coal, cleaned coal, coke, crude oil, gasoline, kerosene, diesel oil, fuel oil, liquefied petroleum gas (LPG) and natural gas. According to the Classification of National Economic Industries (GB/T 4754-2017), the transportation, warehousing and postal service includes railway transportation industry, road transportation industry, ship transportation industry, air transportation industry, pipeline transportation industry, multimodal transport and transport agent industry, handling and warehousing industry and postal industry. The GDP and population data are collected from the China Statistical Yearbook [29]. To eliminate the effect of price fluctuations, GDP and GDP per capita values are converted into the constant price in 2016 from the current prices using the consumer price index (CPI). Due to data limitations in Taiwan, Hong Kong, Macao and Tibet, these areas are excluded in this study. Data of Chongqing is from 1997 to 2016 since it was established in 1997. Data of Fujian in 1997-1998 and Ningxia in 2000-2002 are absent in the statistical yearbook.”

The supplement content in the conculsion section is as follows:

Due to the limitation of energy consumption statistics in the transport sector in China, a certain uncertainty exists in this study. Although the uncertainty exists, it won’t significantly influence the regularities and results since the main aim of this study is to explore the spatiotemporal pattern and uncover its decoupling status from economic growth. With the continuous improvements in the energy consumption statistics in China, more accurate results of the transport CO2 emissions can be obtained in the future. In addition, the identification of the driving forces of the decoupling of transport CO2 emissions from economic growth is of great significance to achieve strong absolute decoupling state. Therefore, identifying influencing factors of decoupling transport CO2 emissions from economic growth and quantifying theirs effects on decoupling status will be an important topic in the next step.”

ü  It is important for identifying driving forces.

Thank you so much for your suggestions about further identifying driving forces. Economic growth is one of the most important driving forces of CO2 emissions in the transport sector. This manuscript focused on identifying the decoupling statues of transport CO2 emissions from economic growth since the economic growth is desirable, while the CO2 emissions are not. In addition, the identification of driving forces of the decoupling of transport CO2 emissions from economic growth is also an important and interesting topic. All authors would like to further explore the driving forces of the decoupling status in the next step of our research. We have added the discussion content in the Conclusion section of this manuscript.

The supplementary content is as follows:

“Due to the limitation of energy consumption statistics in the transport sector in China, a certain uncertainty exists in this study. Although the uncertainty exists, it won’t significantly influence the regularities and results since the main aim of this study is to explore the spatiotemporal pattern and uncover its decoupling status from economic growth. With the continuous improvements in the energy consumption statistics in China, more accurate results of the transport CO2 emissions can be obtained in the future. In addition, the identification of the driving forces of the decoupling of transport CO2 emissions from economic growth is of great significance to achieve strong absolute decoupling state. Therefore, identifying influencing factors of decoupling transport CO2 emissions from economic growth and quantifying theirs effects on decoupling status will be an important topic in the next step.

ü  Figure 2 is confusing.

Thank you for your comments. Figure 2 in this manuscript shows the temporal trends of national CO2 emissions’ proportion the transportation sector derived from ten fuel categories. Since the CO2 emissions derived from the top five categories fuel are much higher than the others, if we put all the CO2 emission proportion derived from the ten categories fuel in one graph, the curves of the proportion of CO2 emission derived from the lower five categories fuel will be very closed to the X-axis and can’t be observed clearly.

The above is our modification. Thank you again for your warm hearted guidance. We hope that our revised manuscript can meet your approval. We have tried our best to improve our manuscript and made some corrections in the manuscript. These corrections will not influence the main content of this paper. We appreciate for Editors/Reviewers’ warm work earnestly, and hope that the correction will meet with approval. If you have any question, please don’t hesitate to contact us. Thank you very much!

Sincerely,

Suocheng Dong, Yu Li

Reviewer 3 Report

Dear authors,

This research shows the efforts China is making to combat negative externalities in the transport sector. Although much remains to be done, the most important thing is that this type of research helps to create new measures to combat climate change.

For all these reasons, I have made a number of proposals so that this article can be referenced as initial research for the transport sector in China.

I think the research that has been done is very interesting.  I believe that those responsible for the design of environmental policies can obtain very significant data in order to be able to design new measures that help to reduce CO2 emissions.

Contributions for improvement the research

 NEW Keywords; Climate change, Indicator of Deocupling, transition to low carbón economy,negative externalities of the transport sector

In my opinion, the authors should include a contextualization about the united nations objectives on climate change, also some data from the IPCC intergovernmental panel, etc., with the objective of strengthening the importance of this research

Also I have considered to indicate some articles that have relation with the topic of the study so that the authors can read them and obtain ideas to improve different sections of this article, especially I refer to the section of Introduction, Results and discussion and Conclusions.

Introduction

O.D.S ( https://www.un.org/sustainabledevelopment/sustainable-development-goals/)

Huijts, N.M.; de Vries, G.; Molin, E.J. A positive Shift in the Public Acceptability of a Low-Carbon Energy Project After Implementation: The Case of a Hydrogen Fuel Station. Sustainability 2019, 11, 2220.

Ren, F.-R.; Tian, Z.; Shen, Y.-T.; Chiu, Y.-H.; Lin, T.-Y. Energy, CO2, and AQI Efficiency and Improvement of the Yangtze River Economic Belt. Energies 2019, 12, 647. 

WarbWarbroek, B.;  Hoppe, T.; Coenen, F.; Bressers, H. The Role of Intermediaries in Supporting Local Low-Carbon Energy Initiatives. Sustainability 2018, 10, 2450

Quesada-Rubio, J.M.; Villar-Rubio, E.; Mondéjar-Jiménez, J.; Molina-Moreno, V. Carbon dioxide emissions vs. allocation rights: Spanish case analysis. Int. J. Environ. Res. 2011, 5, 469–474.

Zhu, L.; He, L.; Shang, P.; Zhang, Y.; Ma, X. Influencing Factors and Scenario Forecasts of Carbon Emissions of the Chinese Power Industry: Based on a Generalized Divisia Index Model and Monte Carlo Simulation. Energies 2018, 11, 2398.

Conclusions

Villar-Rubio, E.; Quesada Rubio, J.M.; Molina Moreno, V. Convergence Analysis of Environmental Fiscal Pressure Acrosss EU-15 Countries. Energy Environ. 2015, 26, 789–802.

University of Adelaide. (2019, March 20). IPCC is underselling climate change. ScienceDaily. Retrieved April 14, 2019 from www.sciencedaily.com/releases/2019/03/190320102010.htm

Zhu, L.; He, L.; Shang, P.; Zhang, Y.; Ma, X. Influencing Factors and Scenario Forecasts of Carbon Emissions of the Chinese Power Industry: Based on a Generalized Divisia Index Model and Monte Carlo Simulation. Energies 2018, 11, 2398.

Soheila Khoshnevis Yazdi, Anahita Golestani Dariani.  (2019) CO2 emissions, urbanisation and economic growth: evidence from Asian countries. Economic Research-Ekonomska Istraživanja 32:1, pages 510-530.

Author Response

Response to the reviewers’ comments:

Dear authors,

This research shows the efforts China is making to combat negative externalities in the transport sector. Although much remains to be done, the most important thing is that this type of research helps to create new measures to combat climate change. For all these reasons, I have made a number of proposals so that this article can be referenced as initial research for the transport sector in China. I think the research that has been done is very interesting. I believe that those responsible for the design of environmental policies can obtain very significant data in order to be able to design new measures that help to reduce CO2 emissions.

Thank you very much for your guidance and encouragement of our manuscript entitled “The spatiotemporal pattern of decoupling transport CO2 emissions from economic growth across 30 provinces in China” (Manuscript ID: sustainability-491083). Your comments enlighten us a lot. We have read the research papers you list for us carefully. Some of them are referred in the revised version of our manuscript. We have also made detailed revisions in our revised manuscript according to your suggestions. We hope that the revised manuscript can meet your approval. Revised portion are marked in the manuscript. The detail (point-by-point) list of the response is as follows:

ü  Contributions for improvement the research: NEW Keywords: Climate change, Indicator of Decoupling, transition to low carbon economy, negative externalities of the transport sector

Thank you for your suggestions on the key words of our manuscript. The new key words you suggested make the key words section more comprehensive and complete. We have revised the key words according to your suggestion.

ü  In my opinion, the authors should include a contextualization about the United Nations objectives on climate change, also some data from the IPCC intergovernmental panel, etc., with the objective of strengthening the importance of this research.

Thank you very much for your precious advice on the Introduction section. According to your suggestion, we have supplied the sustainable development goal 13 ‘climate action’ proposed by the United Nations in the contextualization of introduction. The data related to transport energy consumption and CO2 emission from International Energy Agency are also added to highlight the importance of the research on the decoupling analysis of transport CO2 emissions from economic growth.

The supplementary content in the Introduction section is as follows:

“The reduction of CO2 emissions is a global environmental challenge [1]. The United Nations addressed ‘take urgent action to combat climate change and its impacts’ as one of the goals to achieve sustainable development across the globe [2]. According to the International Energy Agency, the transport sector accounts for 28.81% of total energy consumption in 2017 [3] and 25% of total CO2 emissions in 2016 [4]. The growth of CO2 emissions is led by developing countries experiencing rapid economic growth [5]. In many developing countries, CO2 emissions in the transport sector has attracted great attention because of its high contribution and unprecedented increase in scale and speed [6,7]. China has become the largest country of CO2 emissions since 2005 [8]. The transport sector is a major source of CO2 emissions and the most rapidly growing sector in terms of fuel consumption and CO2 emissions in China [9]. The key challenge for China, as well as other developing countries, is what can be done to reduce CO2 emissions in the transport sector while achieving economic development [10].”

Supplementary References:

2.        United Nations sustainable development goals. Available online: https://www.un.org/sustainabledevelopment/climate-change-2/ (accessed on 18th April 2019)

3.        International Energy Agency (IEA). World Energy Outlook 2018. IEA Publication, Paris, pp. 42.

4.        International Energy Agency (IEA). CO2 Emissions from Fuel Combustion 2018. IEA Publication, Paris, pp. 6.

ü  Also I have considered to indicate some articles that have relation with the topic of the study so that the authors can read them and obtain ideas to improve different sections of this article, especially I refer to the section of Introduction, Results and discussion and Conclusions.

Thank you so much for your detailed provision of the referential research articles for Introduction and Conclusion sections. We have studied these papers carefully and obtain some ideas to improve our manuscript. For example, the sustainable development goals of United Nations are added in the contextualization of introduction; “Ren, F.-R.; Tian, Z.; Shen, Y.-T.; Chiu, Y.-H.; Lin, T.-Y. Energy, CO2, and AQI Efficiency and Improvement of the Yangtze River Economic Belt. Energies 2019, 12, 647” is added in the Introduction section to illustrate the importance of environmental governance in the CO2 emission reduction; “Zhu, L.; He, L.; Shang, P.; Zhang, Y.; Ma, X. Influencing Factors and Scenario Forecasts of Carbon Emissions of the Chinese Power Industry: Based on a Generalized Divisia Index Model and Monte Carlo Simulation. Energies 2018, 11, 2398.” is supplied in the Conclusion section.

The above is our modification. Thank you again for your warm hearted guidance. We hope that our revised manuscript can meet your approval. We have tried our best to improve our manuscript and made some corrections in the manuscript. These corrections will not influence the main content of this paper. We appreciate for Editors/Reviewers’ warm work earnestly, and hope that the correction will meet with approval. If you have any question, please don’t hesitate to contact us. Thank you very much!

Sincerely,

Suocheng Dong, Yu Li

Round 2

Reviewer 1 Report

Thank you for the reviewed manuscript. I see that all my suggestions/comments are taken in to the acount.

Reviewer 3 Report

Dear Authors,

I congratulate you on the final outcome of your investigation. I encourage you to continue working in the same line of research.